# Synthetization and characterization of SnCaAl₂O₃ nanocomposite and using as a superior adsorbent for Pb, Zn, and Cd ions in polluted water

Ali Sayqal[1], Moustafa Gamal Snousy[2], Mahmoud F. Mubarak[3], Ahmed H. Ragab[4], Atef Mohamed Gad Mohamed[5]*, Abeer El Shahawy[6]*

1 Department of Chemistry, Faculty of Applied Science, Umm-Al-Qura University, Makkah, Saudi- Arabia, 2 Egyptian Petroleum Sector, Petrotrade Co., Cairo, Egypt, 3 Petroleum Applications Department, Egyptian Petroleum Research Institute (EPRI), Cairo, Egypt, 4 Department of Chemistry, Faculty of Science, King Khalid University, Abha, Saudi Arabia, 5 Assiut & New Valley Company for Water and Wastewater, Assiut, Egypt, 6 Department of Civil Engineering, Faculty of Engineering, Suez Canal University, Ismailia, Egypt

* abeer_shahawi@eng.suez.edu.eg (AES); atefgad98@yahoo.com (AMGM)

**Data Availability Statement:** All relevant data are within the paper and its Supporting information files.

## Abstract

The presence of heavy metals in drinking water or wastewater poses a serious threat to the ecosystem. Hence, the present study focused on synthesizing SnCaAl₂O₃ core-shell nanoparticles (C.N.P.s) in the α-Alumina phase by thermal annealing a stacked structure sandwiched between two Al₂O₃ layers at low temperatures. The obtained structure showed Sn N.P. floating gate with an Al2O3 dielectric stacked tunneling barrier to remove the excess of these heavy metals from polluted water. To characterize the prepared composites, X-ray diffraction (XRD), field emission scanning electron microscope (FE-SEM), and high-resolution transmission electron microscopy (HR-TEM) were used. The synthesized SnCaAl₂O₃ C.N.P.s composite was examined to utilize it as an adsorbent for removing Zn, Cd, and Pb divalent cations. The removal efficiency was studied by various parameters such as adsorbent dose, pH, contact time, metal concentrations, temperature, and coexisting ions. The experimental results were tested via Langmuir and Freundlich isotherm models. The obtained results were convenient to the Freundlich isotherm model. Moreover, the adsorption thermodynamic behavior of $Zn^{+2}$, $Cd^{+2}$, and $Pb^{+2}$ on the synthesized composite was examined, and the process is endothermic and spontaneous under experimental conditions. The results illustrated that the adsorption efficiency of the SnCaAl₂O₃ core-shell nanoparticles (C.N.P.s) ranged from 88% to about 100% for all cations.

## 1. Introduction

Water pollution. by toxic heavy metal ions is a critical environmental problem that may pose serious health effects. The dissolved metal ions do not undergo decomposition in nature, resulting in significant health hazards to humans and the ecosystem [1–4]. Mining and

**Funding:** The authors would like to thank the Dean of Science and Research at King Khalid University for giving financial support via the General Research Project: grant no. (R.G.P.1/28/43), Saudi Arabia. And The authors would like to thank the Deanship of Scientific Research at Umm Al-Qura University for supporting this work by Grant Code: (22UQU4280446DSR02).

**Competing interests:** The authors have declared that no competing interests exist.

smelting operations, industrial production/usage, residential wastes, sewage discharge, and agriculture are the primary sources of heavy metals in the environment [5–8].

Among heavy metals, $Zn^{+2}$, $Cd^{+2}$, and $Pb^{+2}$ ions are commonly available in industrial wastewater [9]. Biogeochemical cycles introduce these elements into the food chain as toxins. Causes highly poisonous and tends to concentrate in muscle and fatty tissues. Heavy metals can pass through the water stream, air, and soil for long distances, so it is too difficult to assess their impact on the environment [10]. Heavy metals can stand stable for many years without decomposition resulting in various risks [11–13]. Heavy metals removal from wastewater can be applied using different techniques such as ion exchange, reverse osmosis, separation with flotation, adsorption, and absorption.

The common removal system is the adsorption of ions on the surface of many solid materials such as clay, zeolites, activated charcoal, or silica gel. These solid materials have suitable properties, such as large pore volumes, large surface area, high porosity, and high exchange capacity for cations [14, 15]. Supercapacitor electrode materials that include transition metal oxides, phosphides, hydroxides, conductive polymers, and layers of hydroxides and selenides are commonly employed [16–18]. But these different adsorbent materials cannot produce large amounts of treated water. They must be used in huge quantities of these materials to obtain the required quantities of treated water [19–22]. It is known that nanoparticles of calcium oxide (CaO) and ($CaAl_2O_3$) appear to have unique properties due to their high adsorption capacity and their high catalytic activity [23]. On another side, the nanocomposites of $SnAl_2O_3$ are characterized by a large surface area due to the encasement of Al in the $SnO_2$ lattice, which would generate an interstitially solid solution, causing composite surface area to increase [24].

This work aims to manufacture $SnCaAl_2O_3$ core-shell nanoparticles, and this material is $SnAl_2O_3$ as a strong adsorbent and ($CaAl_2O_3$) as a potent catalyst. So, $SnCaAl_2O_3$ composite has superior efficiency and can adsorb heavy metal cations. Furthermore, the absorption properties and absorption mechanisms of the prepared composite $SnCaAl_2O_3$ were studied to remove $Zn^{+2}$, $Cd^{+2}$, and $Pb^{+2}$, as well as mixtures of solutions.

## 2. Materials and method

### 2.1. Materials

Zinc chloride heptahydrate ($ZnCl·7H_2O$), lead sulfate pentahydrate ($PbSO_4·5H_2O$), cadmium sulfate heptahydrate ($CdSO_4·7H_2O$), and calcium chloride hexahydrate ($CaCl·6H_2O$) all are analytically pure, (Federal Standard 4528–78), and aluminum nitrate ($Al (NO_3)_3$), chemically pure, Sigma Aldrich, were used to prepare the starting aqueous solutions of CTAB (98% purity, Sigma Aldrich, St. Louis, MO, U.S.A.), $CaCl_2$, $SnCl_2$ (98% purity, Alfa Aesar, Haverhill, MA, U.S.A.), and $NH_4OH$ (25% purity, Union Chemical Works Ltd., Kaohsiung, Taiwan).

### 2.2. Preparation of $Al_2O_3$ nanoparticles

Al2O3 within the SnCaAl2O3 core-shell nanoparticles (C.N.P.s) was prepared using the sol-gel technique [25]. Al(NO3)3 was dissolved in a proper volume of glacial acetic acid with continuous stirring while heating on a hot plate for 3 h at 80˚C. After that, a small volume of polyvinyl alcohol was poured into this solution. The resulting solution was left for another 6 h with the same conditions of stirring and heating [26]. After elapsing this time, the mixture was centrifuged at 3000 rpm for 10 min, washed for many periods with deionized water and ethanol, and dried using an oven dryer at 80˚C for 12 h. Furthermore, the dried powder grounded well and hardened at 700˚C for 3 h in the air using a muffle furnace to produce pure nano Al2O3 layers [27].

## 2.3. Preparation of SnCaAl$_2$O$_3$ Core-Shell Nanoparticles

A known amount of the as-prepared nano Al$_2$O$_3$ shells was first dispersed in 20 mL of deionized water, followed by a pre-determined amount of CTAB, CaCl$_2$, and SnCl$_2$ with stirring and heating at 300 rpm at 40˚C for 2 h, respectively. After mixing well, NH$_4$OH solution was slowly added for pH adjustment of the mixture to reach 10 and left for 12 h [28], then centrifuged at 2000 rpm [29, 30]. The produced solid particles were dried for 24 h at 110˚C and calcined in atmospheric air at a specific temperature for 3 h [31].

## 2.4. SnCaAl$_2$O$_3$ characterization

X-ray diffraction (XRD) spectra for SnCaAl$_2$O$_3$ sample captured via a powder diffractometer (Ultima IV, Rigaku Corp., Tokyo, Japan) and Cu-K radiation. Emission scanning electron microscopy (FE-SEM) with a speeding voltage of 5 kV and a current of 10 A (S-4500, Hitachi Ltd., Tokyo, Japan) and high-resolution transmission electron microscopy (HR-TEM) were used to recognize the morphological structure of the samples. HR-TEM pictures were acquired by dispersing the sample powder in ethanol and depositing it on a copper grid. The FT/IR-4100 spectrometer (Jasco Corp., Hachioji, Japan) was used to generate FT-IR spectra with a resolution of 2.0 cm$^{-1}$ in the wavelength range of 4000–400 cm$^{-1}$.

## 2.5. Heavy metals adsorption by SnCaAl$_2$O$_3$ core-shell nanoparticles

The sorption characteristics of the as-synthesized SnCaAl$_2$O$_3$ C.N.P.s were determined using the technique of limited volume at room temperature (20 ± 2˚). In comparison, samples of SnCaAl$_2$O$_3$ C.N.P.s (S) were stirred in Erlenmeyer flasks with different concentrations of heavy metals (L) ranging from 50 up to 500 mg/L, with the ratio of S:L = 1:5 for 4 h using a lab shaker [32]. The solid adsorbent was then removed from the liquid by centrifugation and filtration. To estimate the Zn$^{+2}$, Cd$^{+2}$, and Pb$^{+2}$ cations contents, analytical atomic absorption spectroscopy (AA-6300, Shimadzu, Kyoto, Japan) was used. The gotten data are the average of three independent experiments. The sorption properties of the used material can be explained using Eq 1, where $Q_{eq}$ (mg$_{eq}$/g) is the static exchange capacity, $C_0$ is the initial concentration, and $C_t$ is the steady-state concentration (mg/L). $V$ is the metal ions volume in solution (L), and $m$ is the sorbent weight (g) [33].

$$Q_{eq} = (C_0 - C_t) \times \frac{V}{m} \qquad (1)$$

In this study, several influences have been examined, such as contact time (10–80 min), pH (2–9), the dosage of adsorbent (0.02 g up to 1 g), metal ions concentration (50–500 mg/L), the effect of temperature (25–66˚C), and effect of coexisting ions Na$^+$ and Ca$^{2+}$.

## 3. Results and discussions

Due to the dangers of heavy metals to the ecosystem, many scientists and researchers have worked hard to find effective and quick solutions to confront these pollutants; [34] examined the use of multiwalled carbon nanotubes (MCNTs) to remove fenuron pesticide from wastewater, [35] used a new method and technique in treating sludge by removing water from sludge using Extracellular polymeric substances (EPS) [36–41].

The present study discussed the novel preparation and characterization of SnCaAl$_2$O$_3$ Nanocomposite material via different characterization techniques. On another side, it examined this new synthetic material as a superior adsorbent to remove heavy metals (Zn$^{+2}$, Cd$^{+2}$, and Pb$^{+2}$) in the wastewater as a new application.

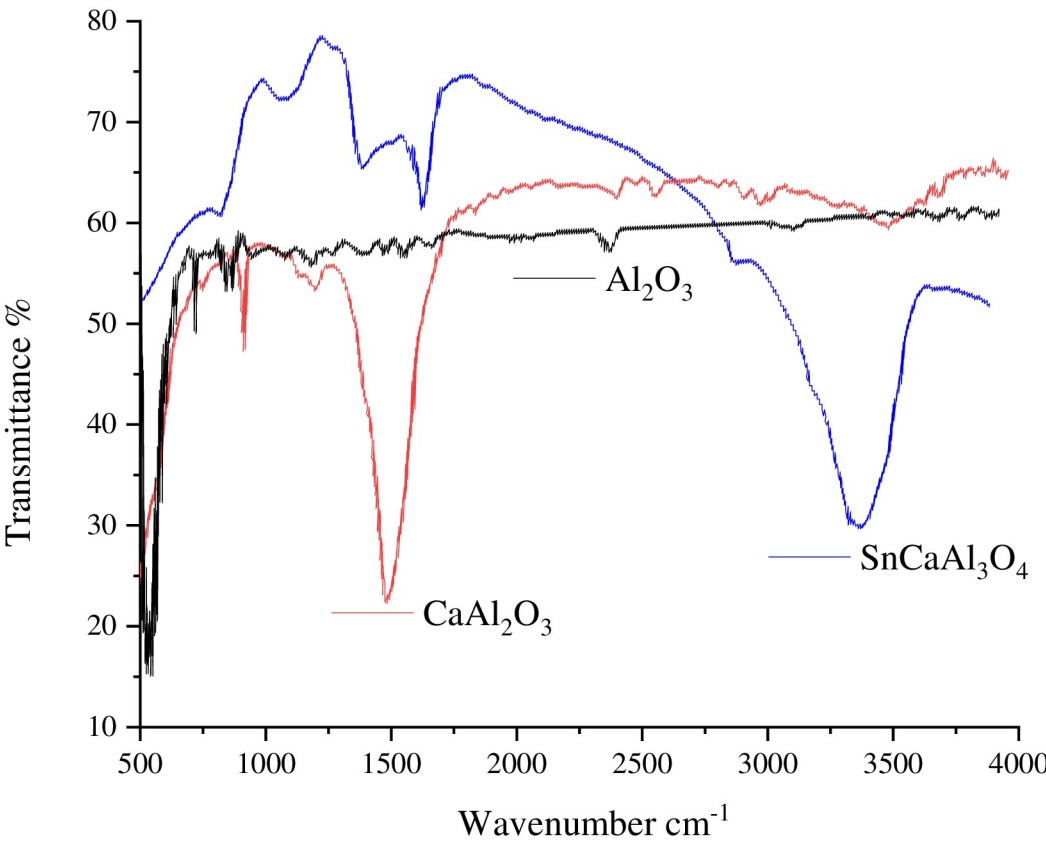

**Fig 1. FTIR of Al₂O₃, CaAl₂O₃, and SnCaAl₂O₃ nanoparticles.**

### 3.1. Alumina nanoparticles characterization

**3.1.1. FTIR spectroscopy.** FTIR spectrum examination was applied to identify the surface functional groups of the prepared nanoparticles (Fig 1). Fig 1 illustrates that the 3457 cm$^{-1}$ peak is related to the O–H group expansion vibration related to the Al–O.H. framework. The 1629 cm$^{-1}$ peak is related to the water molecules which are adsorbed physically at the adsorbent surface [42]. In contrast, the wavelength-wide pattern range of 400–1000 cm$^{-1}$ distinguishes and indicates the creation of the γ-phase alumina [43].

Moreover, the sorption bands of the Al-O-Al bond at 911, 804, and 637 cm$^{-1}$ are indexed to asymmetric/symmetric stretching and bending vibration [24, 44, 45]. The peak below 700 cm$^{-1}$ corresponds to Al$^{+3}$ octahedral arrangement in the SnCa-oxide hcp lattice. At the same time, the peaks between 700 and 950 cm$^{-1}$ represent the C.C.P. lattice tetrahedral sites of the oxide ions occupied with Al$^{+3}$ ions [42]. So, the produced γ-alumina phase coordinates octahedral and tetrahedral units [46].

**3.1.2. Thermal analysis.** TG-DTA analysis investigated the sample's thermal behavior (Fig 2). The sample's T.G. curve showed three weight decreases. Firstly, at 148˚C, the initial sample weight loss was nearly 11%. This weight loss correlates with the D.T.A. profile exothermic peak, representing the desorption of physically adsorbed water. Between 150 and 300˚C, a second large weight loss of nearly 34% is attributable to removing ethanol and other impurities [47]. The last notable weight loss is the third, representing 65% of the total weight loss and occurs at 375˚C, accompanying a strong D.T.A. exothermic peak and is attributed to

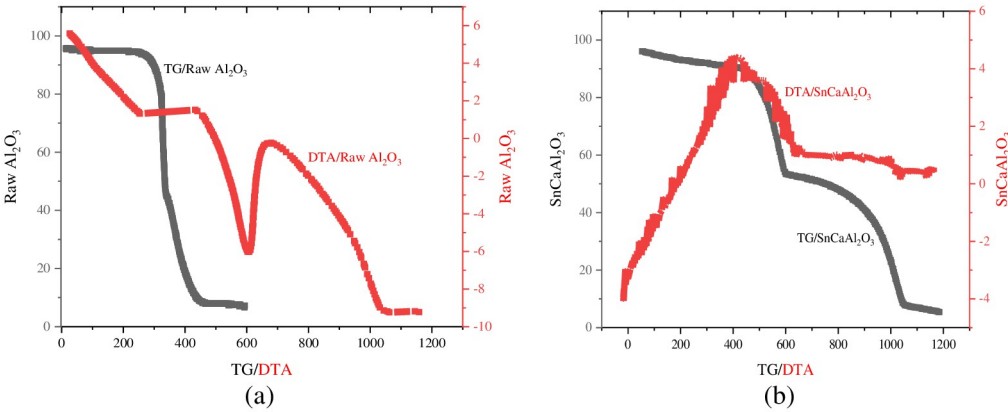

**Fig 2. TG/DTA of (a) Al$_2$O$_3$, and (b) SnCaAl$_2$O$_3$ nanoparticles.**

amorphous aluminum hydroxide dehydration [48]. Whereas there is no considerable weight loss after 600°C, the material is thermally stable. As a result, the sorbent material was calcined and obtained alumina particles phase at this temperature.

**3.1.3. XRD spectroscopy.** XRD examination validated the chemical structure and acquired phase of the as-synthesized nanoparticles. The crystallite size of the as-synthesized materials can be determined using the Scherrer formula (2) and the resultant diffraction peaks [13], in addition to phase confirmation [14]. Fig 3 illustrates the X-ray diffractogram of the formed samples, indicating the creation of the Al$_2$O$_3$ species by the four typical peaks at 2θ of 32°, 37.4°, 45.2°, and 67.5° with comparable reflection planes of 220, 311, 400, and 440. The obtained peaks were nicely matched to the database of standards (JCPDS card 00-029-0063).

$$D = 0.9 \frac{\lambda}{FWHM} \cos \theta \tag{2}$$

Whereas $D$ is the crystallite size in nm, $\lambda$ (0.154056 nm for CuKα radiation) is the wavelength of the monochromatic X-ray beam, $FWHM$ is the full width (rad) for the diffraction peak at half-maximum under consideration, and $\theta$ is the Bragg angle (deg) [44, 49, 50]. The average crystallite size of alumina particles was determined at 55 nm after averaging the results from the (2 2 0), (3 1 1), (4 0 0), and (4 4 0) reflections [51, 52]. On the other hand, XRD results point to the creation of SnCaAl$_2$O$_3$, which has been found in prior investigations in ceramic compounds, such as a combination of Al$_2$O$_3$ and CaCO$_3$, with a formation temperature of around 1100°C. CaAl$_2$O$_4$ was produced at a lower temperature of 250°C in our experiment than previously reported. They discovered that when the CaAl$_2$O$_4$ phase is synthesized from ceramic components (Al$_2$O$_3$ and CaCO$_3$) at high temperatures, it coexists with Ca$_{12}$Al$_{14}$O$_{33}$. However, due to lower temperatures or considerably different beginning circumstances, we could not detect the creation of Ca$_{12}$Al$_{14}$O$_{33}$ in our work. Furthermore, other thermodynamically stable phases, such as CaAl$_{12}$O$_{19}$, Ca$_3$Al$_2$O$_6$, and Ca$_{12}$Al$_{14}$O$_{33}$, could occur in the Al2O3 and CaO system after sintering have not been detected. On the other hand, XRD indicates the production of CaAl$_2$O$_4$ [24, 44, 45].

**3.1.4. Microstructure of prepared nanoparticles.** The morphology and diameter characteristics of the as-prepared nanoparticles can be conducted by HR-TEM analysis. Fig 4 depicts the T.E.M. images of Al$_2$O$_3$ nanoparticles with different hexagonal, cylindrical, and spherical-like structures with average crystallite diameter in the range of 44–55 nm, consistent with those obtained from the XRD analysis [53–55]. For further investigation of the micrograph of

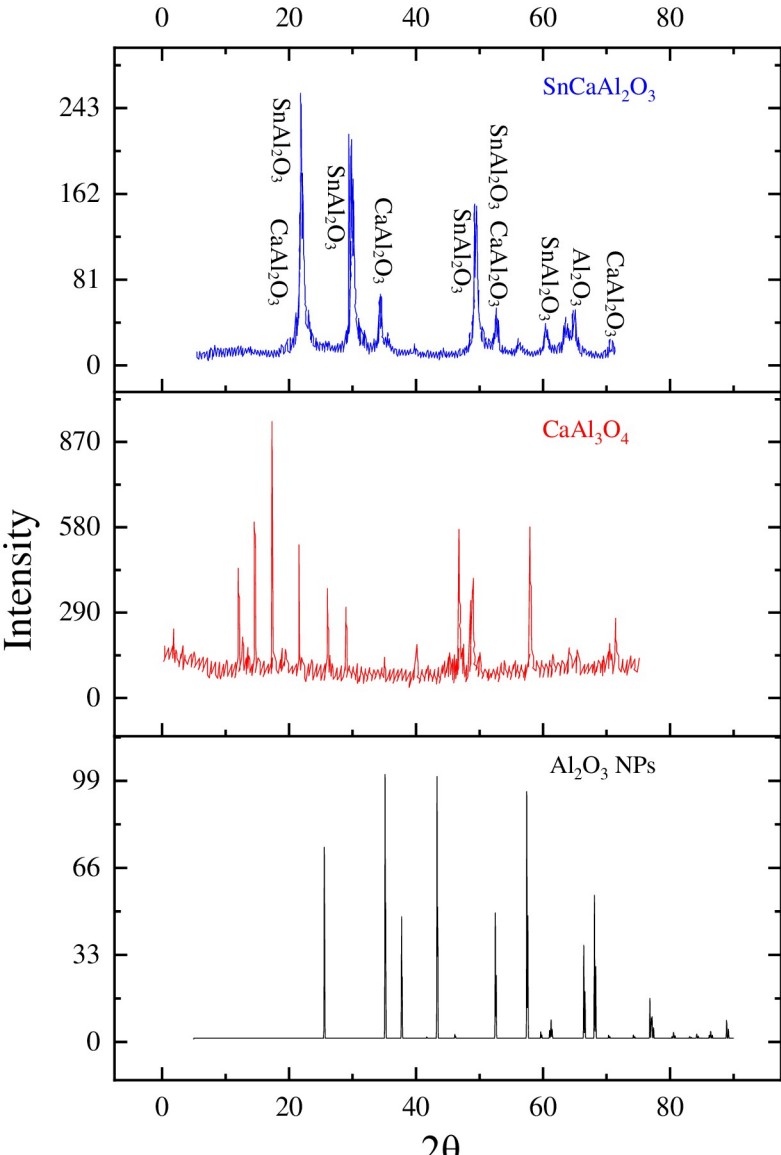

**Fig 3. XRD diffraction patterns of Al$_2$O$_3$ and SnCaAl$_2$O$_3$ nanoparticles.**

SnCaAl$_2$O$_3$ C.N.P.s. It shows a bulk agglomerate structure composed of thin layers of Al$_2$O$_3$ nanoparticles with a Ca-containing layer at the Al$_2$O$_3$ interface. This complicated structure is indexed to the monocalcium aluminate SnCaAl$_2$O$_3$ (JCPDS no. 23–1036).

On the other hand, FE-SEM utilizes for exploring the sample's surface morphology and the S.E.M. images depicted in Fig 5. From Fig 5a, the S.E.M. micrograph of the Al$_2$O$_3$ shows a separate bulk structure composed of thin layers and cracked Al$_2$O$_3$ particles, voids, or holes that can trap Sn and Ca within it. The observed voids could be due to escaping gases during the sample's annealing. Moreover, the surface morphology of the SnCaAl$_2$O nanoparticles shown in Fig 5b has a soft, smooth pattern and sponge shape. As a result of a weak van der Waal bond formation among the particles, they seem to clump together [56].

**3.1.5. Surface area of nanoparticles.** The prepared samples' textural and surface area characteristics were studied using the N$_2$ sorption-desorption technique. The obtained IV

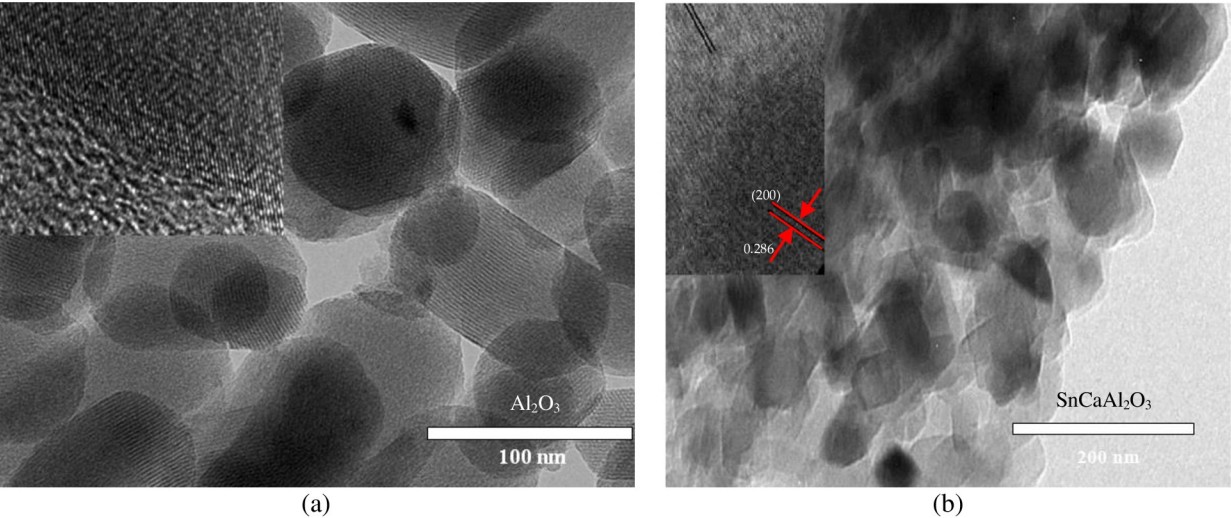

**Fig 4. T.E.M. image of Al₂O₃ (a) and SnCaAl₂O₃ (b) nanoparticles.**

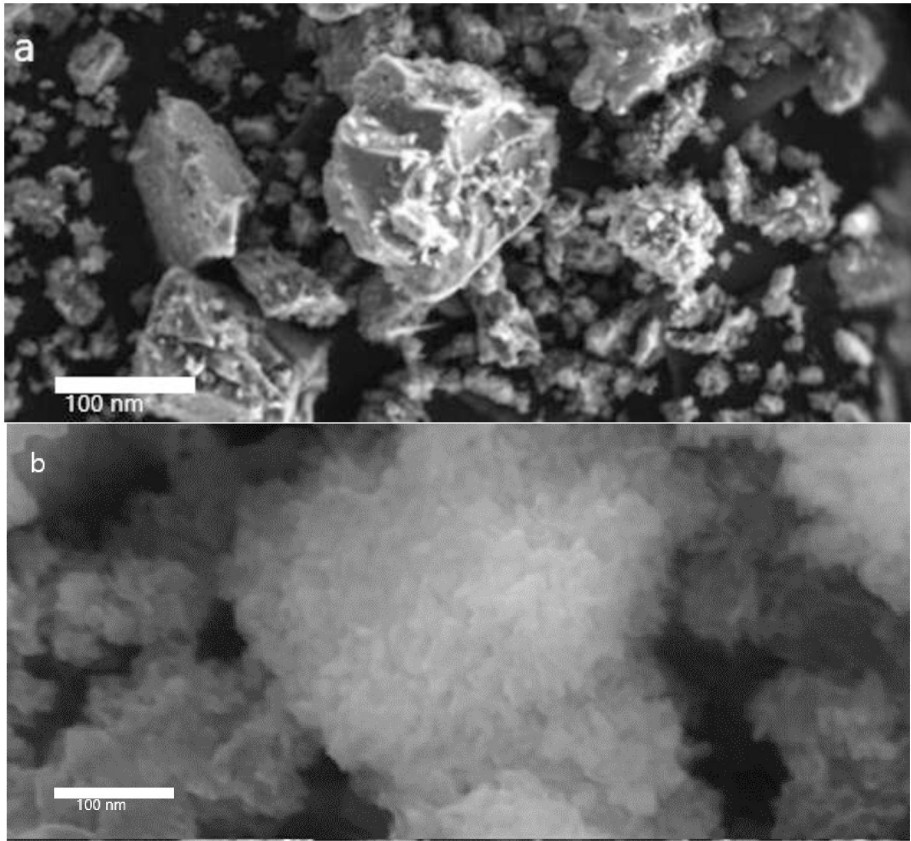

**Fig 5. S.E.M. microstructure of Al₂O₃ (a) and SnCaAl₂O₃ nanoparticles (b).**

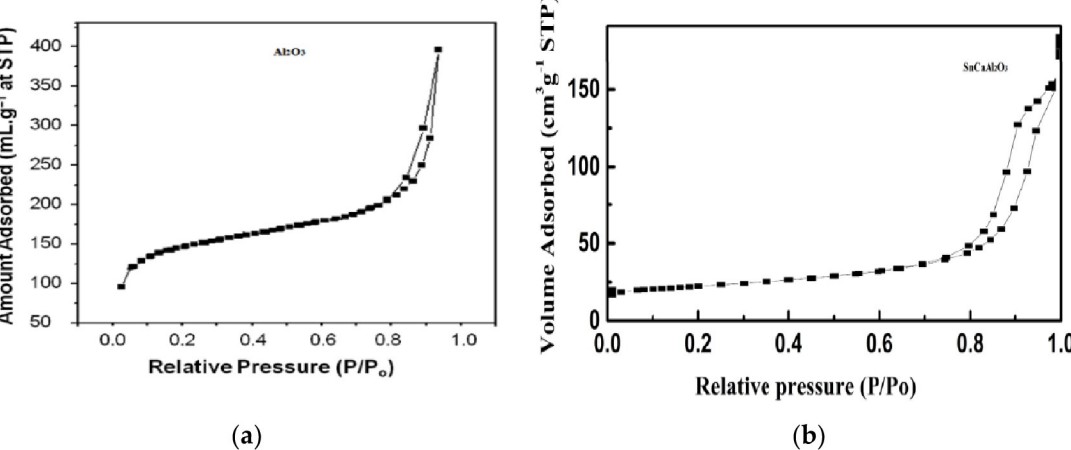

**Fig 6. B.E.T. adsorption-desorption isotherm curve for (a) Al$_2$O$_3$ and (b) SnCaAl$_2$O$_3$.**

isotherm with hysteresis type H1 loop indicates a mesoporous structure [57]. Additionally, the Equation of Brunauer–Emmett–Teller (B.E.T.) [58] was applied to investigate the specific surface area, and it was 108 and 390 m$^2$/g SnCaAl$_2$O$_3$ and Al$_2$O$_3$, respectively (Fig 6). The calculated specific surface area was greater than 344 m$^2$/g, confirming the creation of thermodynamically stable alumina. The B.J.H. pore size distribution curves were analyzed to determine the pore size of the studied samples [59]. As shown in Fig 7, about 90% of the pores were in the range of 2–50 nm, indicating the mesoporous nature of the adsorbent, while the rest (10%) of the pores were below 2 nm, representing the microporous character. The pore

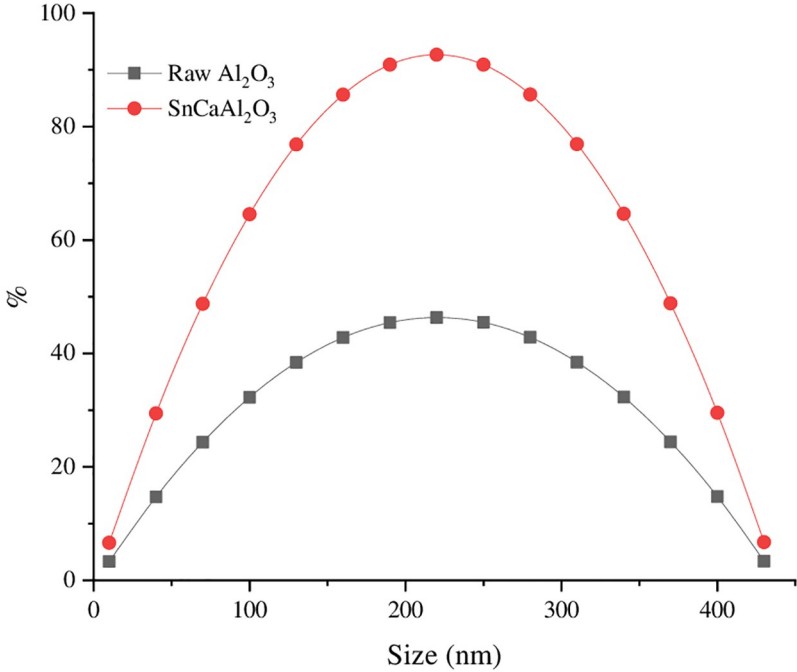

**Fig 7. Particle size distribution of Al$_2$O$_3$ and SnCaAl$_2$O$_3$ NPs.**

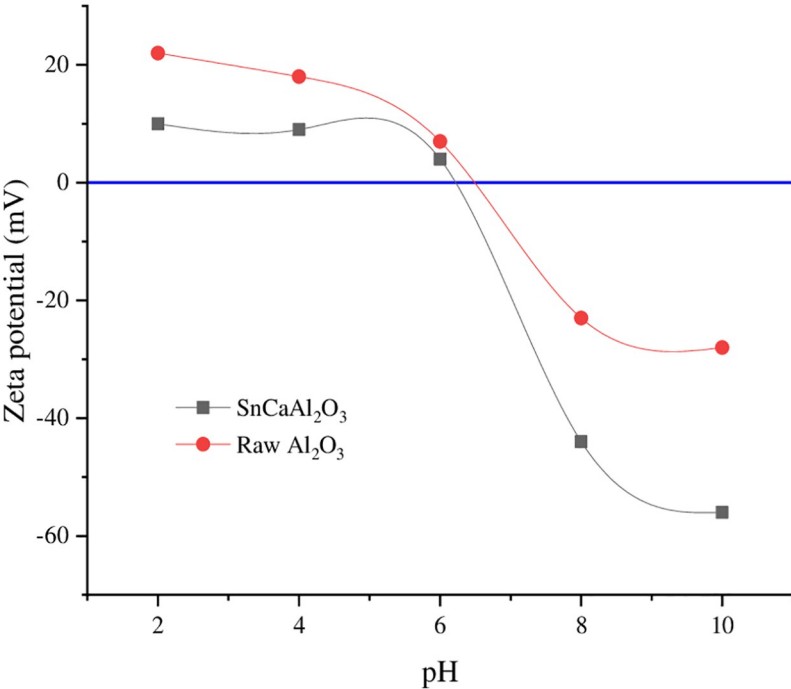

**Fig 8. Zeta potential of Al$_2$O$_3$ and SnCaAl$_2$O$_3$ nanoparticles.**

size and volume of the adsorbent were calculated as 13.6 nm and 0.32 cm$^3$/g, respectively. As a result, mesoporous structures and a large pore volume provide a suitable adsorptive environment for eliminating Zn$^{+2}$, Cd$^{+2}$, and Pb$^{+2}$ ions.

**3.1.6. Zeta optional of the prepared nanoparticles.** The pH of the zero-point charge (pH$_{ZPC}$) is the point at which the surface of the substance is electrically neutral (zero charges), whereas, beneath this point, the material surface is positively charged. In contrast, above this point, it is negatively charged. Knowing the pH of the pH$_{ZPC}$ gives reasonable expectations about the manner of the adsorption process and at which favorable pH range. From Fig 8, the pH$_{ZPC}$ of alumina nanoparticles was 6.4, close to that reported by many authors [60, 61]. The pH$_{ZPC}$ of SnCa is 6.2, indicating the pH$_{ZPC}$ to a lower value due to the defect of oxygen atoms in the composite crystals that make the charge move in a positive direction more than a negative one. In addition, at relatively high N.P.s concentrations, which leads to the particle's aggregation and the effective surface charge on N.P.s decreases, the repulsion between the N.P.s decreases.

## 3.2. Adsorption parameters of heavy metals on SnCaAl$_2$O$_3$ nanoparticles

**3.2.1. pH Effect.** The pH determines the sorption behavior of a solution and whether it will occur in an acidic or alkaline medium. This effect can result in the surface charge of both adsorbent and adsorbate species [26, 60]. Accordingly, the role of solution pH in removing Zn$^{+2}$, Cd$^{+2}$, and Pb$^{+2}$ by SnCaAl$_2$O$_3$ nanoparticles was carried out by pH varying from 2 to 9 at room temperature with an initial Zn$^{+2}$, Cd$^{+2}$, and Pb$^{+2}$ concentration of 500 mg/L. Fig 9 shows that the removal efficiency of Zn$^{+2}$, Cd$^{+2}$, and Pb$^{+2}$ increases gradually from pH 2 until reaching its maximum at pH 7, with the highest removal percent of 84.4%, 87%, and 84.8% for the ions of Zn, Pb, and Cd, respectively. Further increasing the solution pH above 8 may

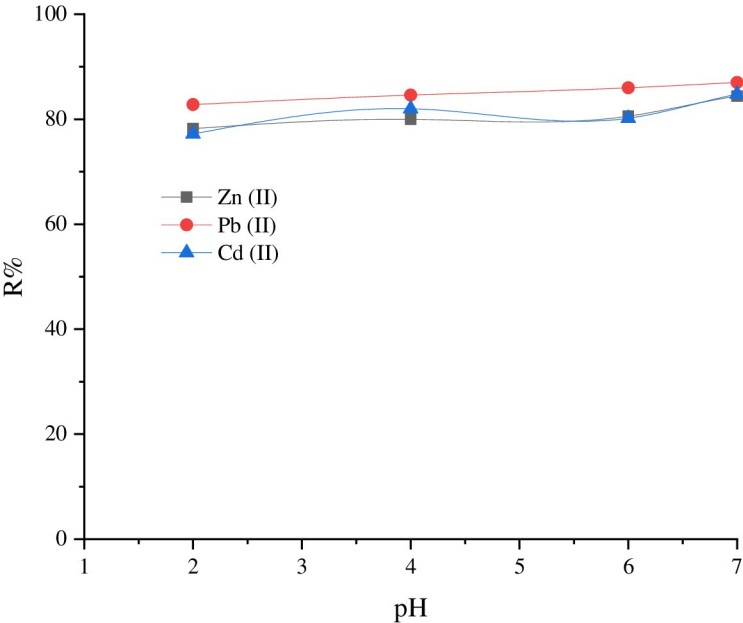

**Fig 9. The pH effect on the SnCaAl$_2$O$_3$ nanoparticles' adsorption of Zn$^{+2}$, Cd$^{+2}$, and Pb$^{+2}$, at constant adsorbent dose, initial concentration of ions, contact time, and volume of polluted water.**

indicate a higher removal percent due to precipitation of the hydroxide form of Zn$^{+2}$, Cd$^{+2}$, and Pb$^{+2}$ ions [62]. The mechanism of the removal process may be explained depending on the surface chemistry of the adsorbent SnCaAl$_2$O$_3$ nanoparticles, where the metal oxide surfaces in an aqueous phase behave amphoterically and respond to pH changes by undergoing acid-base interactions in the aqueous phase [63, 64].

**3.2.2. Adsorbent dosage effect.** The SnCaAl$_2$O$_3$ adsorbent dosage effect on Zn$^{+2}$, Cd$^{+2}$, and Pb$^{+2}$ removal via adding various weights ranging from 0.02 up to 1 g to 50 mL (aqueous solution) of 500 mg/L Zn$^{+2}$, Cd$^{+2}$, and Pb$^{+2}$ as an initial concentration, was investigated. Fig 10 illustrates that the removal percent increases with adsorbent doses from 0.02 to 0.5 g. This may attribute to the superior surface-active sites readily available for adsorption of the pollutant's ions [64]. In addition, further increasing with the added weight of the adsorbent did not result in much increase in the removal percent. In other words, a high dose of the adsorbent may result in agglomeration in the solution, decreasing the removal percent [65–68]. Therefore, the optimum adsorbent dose is 0.5 g.

**3.2.3. Contact time Vs. sorption kinetics.** The effect of contact time on the SnCaAl$_2$O$_3$ nanoadsorbent removal efficiency was examined at pH 8, using a known weight of the adsorbent with a constant high concentration (500 mg/L) of Zn$^{+2}$, Cd$^{+2}$, and Pb$^{+2}$. Fig 11 represents the removal percent against the stirring time between 10 and 80 min. It is recognized that as the contact period grew until 30 min, sorption increased. Then, it reached equilibrium [69]. The equilibrium time needed for SnCaAl$_2$O$_3$ NPs interactions with all initial Zn$^{+2}$, Cd$^{+2}$, and Pb$^{+2}$ concentrations in a steady state is 30 min. It indicates that the interaction is concentration-independent. The fast removal of the studied ions within 30 min may interrelate to the adsorbent surface free-active sites for attracting the pollutants ions from the aqueous solution [70]. Further increasing the interaction time, the removal rate decreases slowly. The adsorption curves show almost straight lines corresponding to the full coverage and saturation of the adsorbent surface-active sites (Fig 11C).

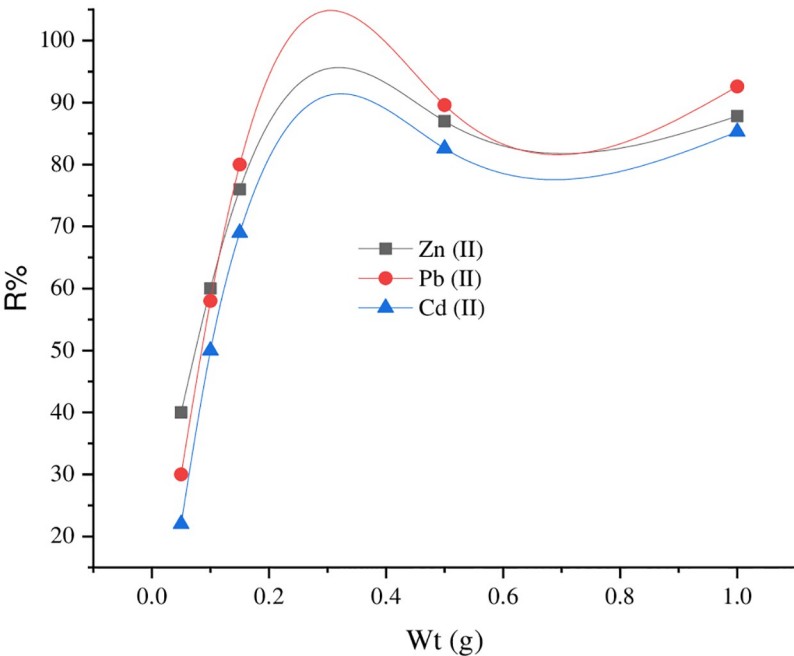

**Fig 10. The SnCaAl$_2$O$_3$ nanoparticles dose effect on the adsorptions of Zn$^{+2}$, Cd$^{+2}$, and Pb$^{+2}$, at the constant of volume solution (50 ml) and initial concentration of these ions (500 mg/l).**

### 3.3. Adsorption kinetics and isotherm models

**3.3.1. Kinetics models.** The heavy metals sorption mechanism was studied using SnCaAl$_2$O$_3$ nanoparticles, and kinetics models were used to examine the experimental data obtained in the study. Pseudo-first-order and pseudo-second-order kinetic models were used. Adsorption kinetic parameters were studied for contact times varied from 5 to 200 min via monitoring the adsorption capacity of Zn$^{+2}$, Pb$^{+2}$, and Cd$^{+2}$ by the adsorbent through this time interval.

Pseudo-First-Order Kinetics (Lagergren)

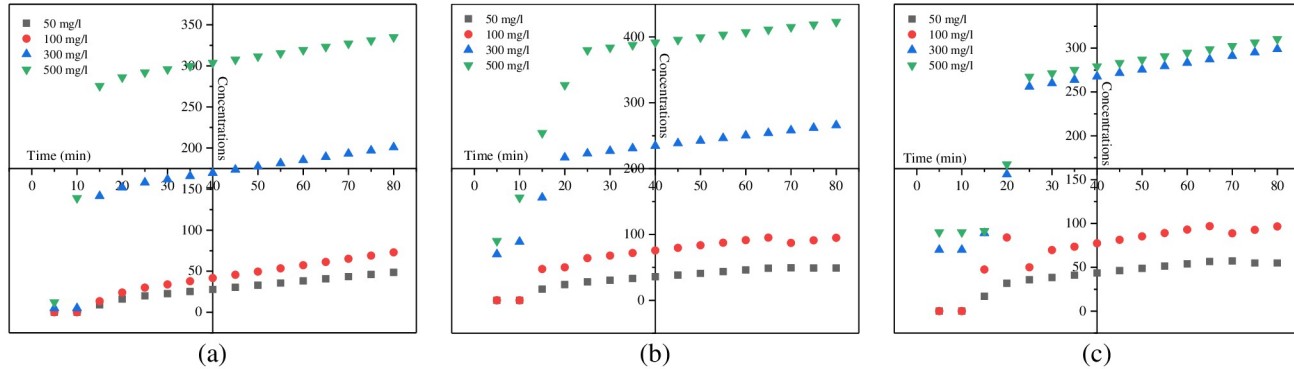

**Fig 11. Influence of contact time on adsorption of (a) Zn$^{+2}$, (b) Pb$^{+2}$, and (c) Cd$^{+2}$ in SnCaAl$_2$O$_3$ NPs at constant adsorbent dose, initial concentration of ions, and volume of polluted water.**

**Table 1. The pseudo-first-order and pseudo-second-order kinetic parameters for sorption of Zn$^{+2}$, Pb$^{+2}$, and Cd$^{+2}$ using SnCaAl$_2$O$_3$ nanoadsorbent.**

| Pseudo Order | Ions | | |
|---|---|---|---|
| Kinetics Model of Pseudo-First-Order | Zn(II) | Pb(II) | Cd(II) |
| Qe, mg/g | 46.27510 | 35.38621 | 40.32178 |
| K$_1$ | −0.0096 | −0.02261 | −0.02735 |
| $R^2$ | 0.933421 | 0.891272 | 0.922456 |
| Kinetics Model of Pseudo-Second-Order | Zn(II) | Pb(II) | Cd(II) |
| Qe, mg/g | 50 | 40.87709 | 43.48520 |
| K$_2$ | 0.134156 | 0.180427 | 0.278443 |
| $R^2$ | 0.995548 | 0.993413 | 0.994842 |

Eq (3) gives the linear form:

$$\log(qe - qt) = \log(qe) - \mathrm{K}_1 2.303t \tag{3}$$

Table 1 reveals the parameters of the Lagergren model created from the experimental kinetic data. The straight-line emerges through plotting log(qe – qt) against t. Although, K1 and the theoretical qe may be derived from slope and intercept (S1 Fig). The plot of $\log(q_e - q_t)$ illustrates the respective estimated value against the corresponding equilibration time $t$ in hours.

Eq (4) describes Pseudo-second-order model [71, 72].

$$t_{qt} = 1\mathrm{K}_2 q_e 2 + t_{qe} \tag{4}$$

Whereas $q_e$ and $q_t$ are the amount of metal ions adsorbed at equilibrium time and instant time ($t$). K$_1$ = adsorption rate constant (g/mg min) and K$_2$ = pseudo-second-order reaction rate constant (g/mg min). The graphs of kinetic models for Zn$^{+2}$, Pb$^{+2}$, and Cd$^{+2}$ adsorption using SnCaAl$_2$O$_3$ nanoadsorbent are stated in (S1 Fig).

Table 1 reveals that the sorption data fit was consistent with the pseudo-second-order model by a strong correlation coefficient $R^2$ and is near 1 for Zn$^{+2}$, Pb$^{+2}$, and Cd$^{+2}$. Furthermore, the pseudo-second-order model's Qe$_{cal}$ (calculated) value is very equivalent to the Qe$_{exp}$ (experimental), which is considered after Eq (2) through a correlation coefficient greater than the pseudo-first-order. These results suggested the involvement of chemisorption in Zn$^{+2}$, Pb$^{+2}$, and Cd$^{+2}$. Chemisorption synchronization arises once a chemical bonding originates between metals and adsorbent surface that increases with active sites.

**3.3.2. Isotherm models.** Table 2 reveals that the isotherm of SnCaAl$_2$O$_3$ nanoparticle's multilayer adsorption was acceptable for the Freundlich model. The $q_e$ fitted value via the Langmuir model differs from the current data. Cations are built through cross whiskers and the surface of whiskers and may accumulate in the SnCaAl$_2$O$_3$ NPs pores [64, 65]. The cation exchange and surface adsorption process may describe the adsorption mechanism. Therefore, the adsorption of the heavy metals on SnCaAl$_2$O$_3$ nanoparticles may comprise both single layer chemical adsorption and multilayer physical adsorption. Hence, the features of isotherms display multilayer physical adsorption (S2 Fig).

**3.3.3. Heavy metals concentration Vs. adsorption equilibrium.** The adsorption capacity of the as-synthesized SnCaAl$_2$O$_3$ nanoparticles was investigated at pH = 8 with different concentrations of Zn$^{+2}$, Pb$^{+2}$, and Cd$^{+2}$ in the range of 50 to 500 mg/L, with 30 min stirring time at 25°C. Fig 12 revealed that the removal efficiency increased at low initial pollutants concentration and decreased gradually with increasing the concentration at 500 mg/L. This is related

**Table 2. Constants of Langmuir, Freundlich, isotherm.**

| Langmuir | | Freundlich | |
|---|---|---|---|
| Pb | | | |
| b | 0.020293 | $K_f$ | 1.393201 |
| $R^2$ | 0.991001 | 1/n | 0.530594 |
| $R_L$ | 0.985558 | $R^2$ | 0.961846 |
| $q_{max}$ | 36.61836 | - | - |
| Cd | | | |
| b | 0.013614 | $K_f$ | 1.01943 |
| $R^2$ | 0.995437 | 1/n | 0.992335 |
| $R_L$ | 1.469115 | $R^2$ | 0.999525 |
| $q_{max}$ | 40.00987 | - | - |
| Zn | | | |
| b | 0.011401 | $K_f$ | 1 |
| $R^2$ | 0.929377 | 1/n | 1 |
| $R_L$ | 1.754287 | $R^2$ | 1 |
| $q_{max}$ | 34.60915 | - | - |

to the adsorbent free active site's saturation. In comparison, the adsorption capacity of the adsorbent increased in higher initial concentrations of the pollutants because the driving force increased, which increased the interactions between metal ions and the active adsorbent sites [71, 73, 74]. Fig 12 reveals that the adsorption capacity Qe displays a reverse relationship to the initial concentration *Co*. In contrast, metal ions have a strong negative correlation as the initial concentration *Co* increases.

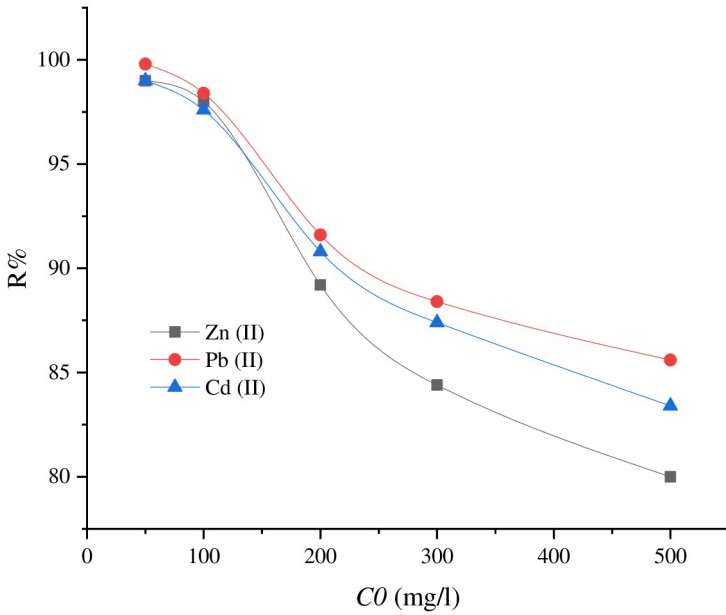

**Fig 12. Adsorption capacities of SnCaAl₂O₃ nanoparticles as a function of Zn⁺², Pb⁺², and Cd⁺² ions concentration at constant adsorbent dose, contact time, and volume of polluted water.**

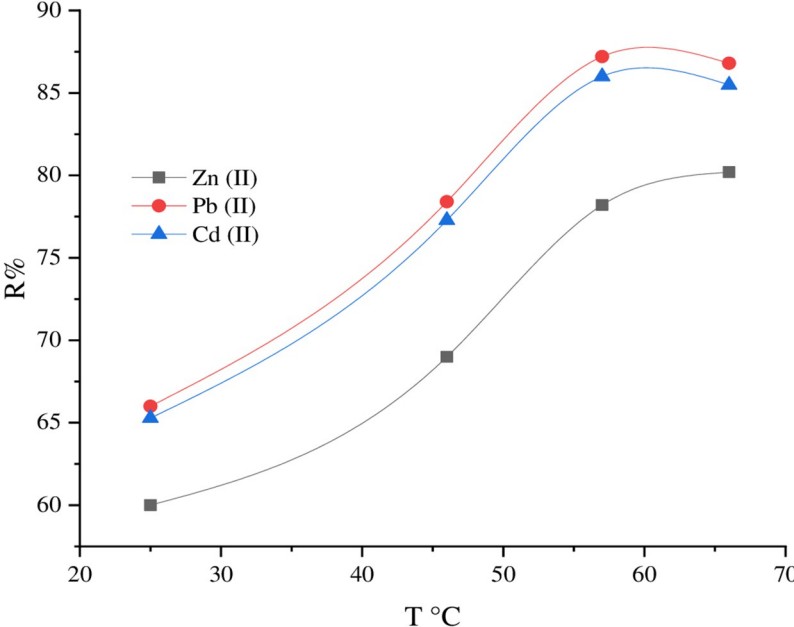

**Fig 13. The Zn$^{+2}$, Pb$^{+2}$, and Cd$^{+2}$ adsorption by the SnCaAl$_2$O$_3$ nanoparticles under various temperatures (25–66˚C), at constant adsorbent dose, contact time, optimum pH, initial concentration of ions, and volume of polluted water.**

**3.3.4. The role of temperature on the adsorption rate of metal ions.** The effect of the adsorption rate of Zn$^{+2}$, Pb$^{+2}$, and Cd$^{+2}$ by the SnCaAl$_2$O$_3$ nanoparticles was considered by mixing 1 g of SnCaAl$_2$O$_3$ with 15 mL of metal ions solution with a concentration of 500 mg/L accompanied by shaking for 30 min in a temperature range of 25–66˚C. Fig 15 displays that the removal efficacy of the pollutants increased with the temperature increase, which confirms the endothermic nature of the adsorption method. This behavior may result from the increase in the speed or the mobility of the pollutants in the solution toward the active adsorbent sites [75, 76]. As shown in Fig 13, the adsorption capacity (Qe) is proportional directly to the elevated temperature.

## 3.4. Thermodynamics study

Thermodynamic parameters ΔG (Standard Gibbs free energy), ΔH (standard enthalpy), and ΔS (standard entropy) are considered for Zn$^{+2}$, Pb$^{+2}$, and Cd$^{+2}$ adsorption onto SnCaAl$_2$O$_3$ nanoparticles, the Eqs (5–7) calculate ΔG, ΔH, and ΔS [69]:

$$\Delta G^\circ = RT \ln K_L^{'} \tag{5}$$

$$\ln \frac{K_{L2}}{K_{L1}} = -\left(\frac{\Delta H^\circ}{R}\right)\left(\frac{T1 - T2}{T_1 T_2}\right) \tag{6}$$

$$\Delta S^\circ = \frac{\Delta H^\circ - \Delta G^\circ}{T} \tag{7}$$

The Langmuir constants are $K_{L1}$ and $K_{L2}$ at $T_1$ and $T_2$, respectively. As well as, R is the gas constant (8.314 J mol$^{-1}$ K$^{-1}$).

**Table 3. ΔH, ΔS, and ΔG for Zn$^{+2}$, Pb$^{+2}$, and Cd$^{+2}$ adsorption on the SnCaAl$_2$O$_3$ nanoparticles.**

| Pb | | | |
|---|---|---|---|
| T (K) | H (J/mol) | S (J/mol K) | G (J/mol) |
| 298.15 | −15448 | 58.73833 | −32960.8 |
| 313.15 | | | −2288.52 |
| 333.15 | | | −5746.57 |
| 353.15 | | | −4480.29 |
| Cd | | | |
| T (K) | H (J/mol) | S (J/mol K) | G (J/mol) |
| 298.15 | −11575.4 | 50.04279 | −26495.7 |
| 313.15 | | | −28503.1 |
| 333.15 | | | −31995.8 |
| 353.15 | | | −31988.9 |
| Zn | | | |
| T (K) | H (J/mol) | S (J/mol K) | G (J/mol) |
| 298.15 | −7924.8 | 41.09136 | −20176.2 |
| 313.15 | | | −22078.1 |
| 333.15 | | | −22899.9 |
| 353.15 | | | −23721.8 |

Table 3 shows the thermodynamic parameters of Zn$^{+2}$, Pb$^{+2}$, and Cd$^{+2}$ adsorption onto SnCaAl$_2$O$_3$ nanoparticles at varied temperatures. The ΔH and ΔS changes were calculated as the slope and intercept of the linear plot ln$Kc$ against $1/T$ (S3 Fig) [77]. From the Van't Hoff equation ΔH, ΔS, and ΔG are calculated using Eqs (5), (6) and (7). The negative values were revealing of a spontaneous adsorption process as temperature increases. The positive values refer to the affinity of the SnCaAl$_2$O$_3$ nanoparticles for Zn$^{+2}$, Pb$^{+2}$, and Cd$^{+2}$. Therefore, chemical and physical adsorption may occur simultaneously [70]. The adsorption heat for van Edward force is in the range of 4–10, the hydrogen bond between 2–40, ligand exchange is 40, dipole interaction ranges from 2–29, and the chemical bond is >60 kJ mol$^{-1}$ [63]. In the current finding, the values are in a similar range, representing the adsorption through the hydrogen bond besides ligand exchange. Thus, physical and chemical adsorption contributed to Zn$^{+2}$, Pb$^{+2}$, and Cd$^{+2}$ adsorption onto SnCaAl$_2$O$_3$ nanoparticles.

### 3.5. Effect of coexisting ions

Many inorganic anions and cations are present in wastewater discharged. Therefore, it is critical to test the adsorbent's selectivity for certain ions in the occurrence of competing ions. Adsorption effectiveness of SnCaAl$_2$O$_3$ nanoparticles for Zn$^{+2}$, Pb$^{+2}$, and Cd$^{+2}$ ions in coexisting ions such as Na$^+$ and Ca$^{2+}$ was investigated. Fig 14 demonstrates that the coexistence of Na$^+$ and Ca$^{2+}$ ions did not affect the removal efficiency of Zn$^{+2}$, Pb$^{+2}$, and Cd$^{+2}$ from SnCaAl$_2$O$_3$ nanoparticles. The strong affinity of SnCaAl$_2$O$_3$ nanoparticles for Zn$^{+2}$, Pb$^{+2}$, and Cd$^{+2}$ may explain via the development of the surface of the outer-sphere complexes on metal hydroxides [78]. When SnCaAl$_2$O$_3$ nanoparticles utilize as adsorbents, Na$^+$ and Ca$^{2+}$ ions in an aqueous solution are not a limiting factor in the treatment process.

### 3.6. Regeneration and desorption

The reusability of SnCaAl$_2$O$_3$ nanoparticles adsorbent for wastewater treatment and removal of the pollutants from the aqueous medium is of great economic value. This property can be

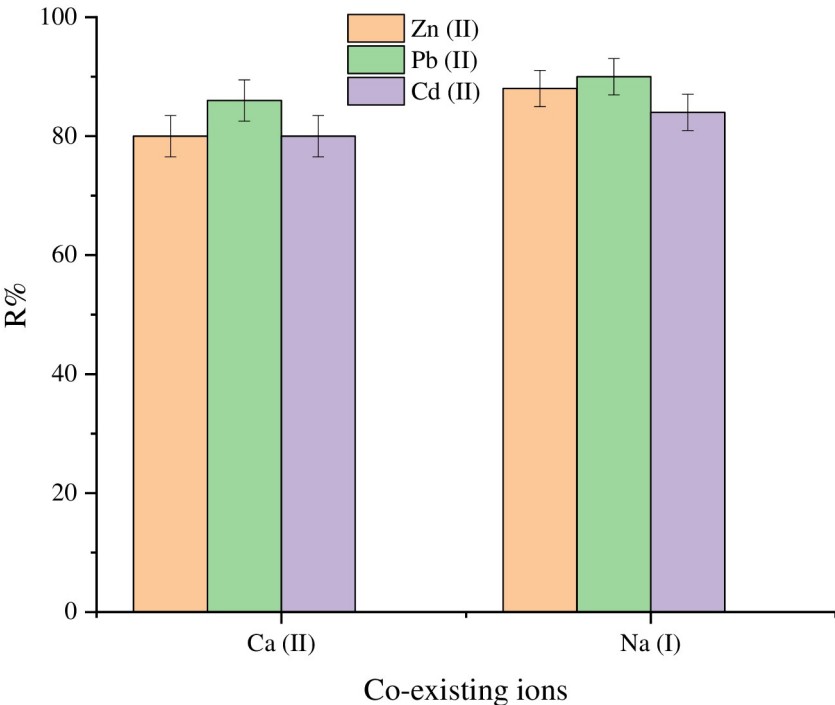

**Fig 14. Coexisting Na$^+$ and Ca$^{2+}$ affect Zn$^{+2}$, Pb$^{+2}$, and Cd$^{+2}$ uptake by SnCaAl$_2$O$_3$ nanoparticles.**

studied by regeneration of the SnCaAl$_2$O$_3$ nanoadsorbent using 30 mL of HCl 0.5 mol L$^{-1}$ and 30 mL of NaOH 0.5 mol L$^{-1}$. The desorption of Zn$^{+2}$, Pb$^{+2}$, and Cd$^{+2}$ can attain via controlling the solution pH. Therefore, NaOH and HCl solutions can regenerate the active adsorbent sites. Therefore, recovery of the adsorption properties of the adsorbent would be achieved [79]. As a result of using the abovementioned concentrations of HCl and NaOH, the desorption efficiencies for Zn$^{+2}$, Pb$^{+2}$, and Cd$^{+2}$ loaded on SnCaAl$_2$O$_3$ nanoadsorbent reached around 100% [80]. This proves that the as-synthesized SnCaAl$_2$O$_3$ nanoadsorbent has suitable adsorption-desorption properties for removing Zn$^{+2}$, Pb$^{+2}$, and Cd$^{+2}$ [81]. As depicted in Fig 15, the adsorption studies were applied with five cycles. The removal efficiencies decreased to 92% and 94% after the third cycle and 90% and 88% after the fifth cycle for Zn$^{+2}$, Pb$^{+2}$, and Cd$^{+2}$ ions, respectively.

## 4. Adsorption mechanism of Zn$^{+2}$, Pb$^{+2}$, and Cd$^{+2}$ by SnCaAl$_2$O$_3$ NPs

Adsorption consists of three main steps: (1) film diffusion; where the motion of the bulk liquid adsorbate is surrounded by a film of the adsorbent, (2) surface adsorption; whereas the adsorbate transport from the film to the surface of the adsorbent, and (3) intraparticle diffusion; the adsorbate transmission to the internal active sites which attach to the heavy metal ions [70].

The slowest step of adsorption (limiting step) is the process that controls the whole adsorption rate. Performed by several kinetic models are applied to examine the rate-limiting step; this includes P.F.O. and PSO. At the same time, the adsorption process is chemisorption's, and the adsorption percentage increase due to the interaction of Zn$^{+2}$, Pb$^{+2}$, and Cd$^{+2}$ ions with the surface functional groups of the adsorbents as a result of the ion exchange mechanism or through hydrogen bonding [69].

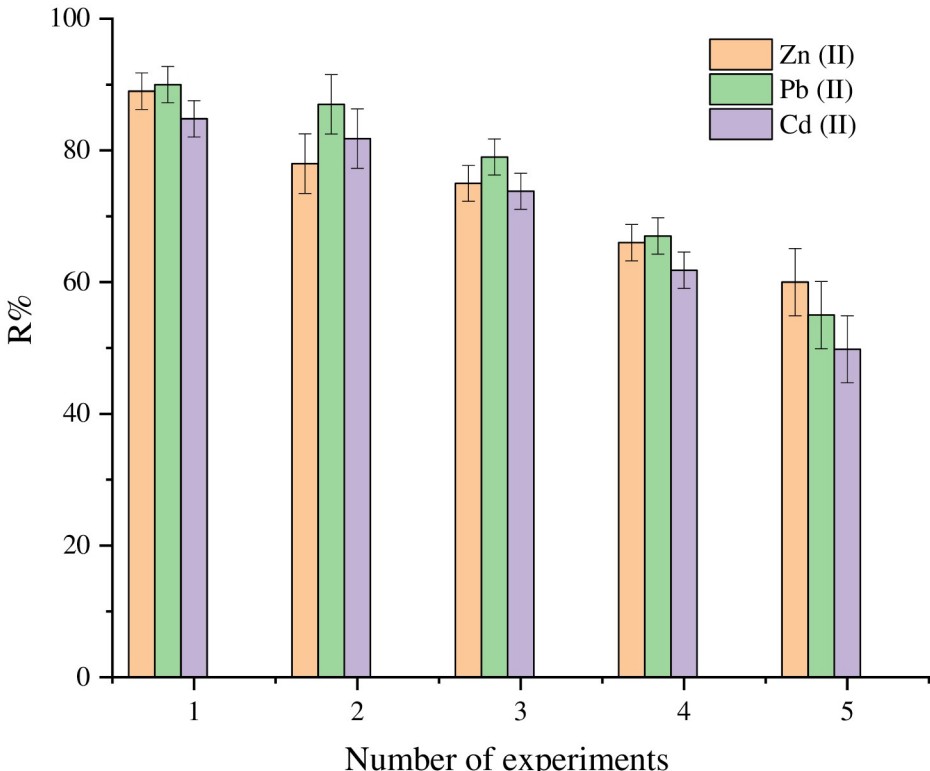

**Fig 15. Desorption of $Zn^{+2}$, $Pb^{+2}$, and $Cd^{+2}$ and regeneration process of $SnCaAl_2O_3$ nanoparticles.**

## 5. Evaluation of the SnCaAl₂O₃ adsorption capacity

Table 4 compares $q_{max}$ values with various adsorbents of $Zn^{+2}$, $Pb^{+2}$, and $Cd^{+2}$, the $SnCaAl_2O_3$ nanoparticles existing advanced adsorption capacity. On the other hand, $SnCaAl_2O_3$ nanoparticles have a reasonable material cost and economic benefit as promising materials for alleviating $Zn^{+2}$, $Pb^{+2}$, and $Cd^{+2}$.

## 6. Conclusions

The present study focused on synthesizing and characterizing a novel $SnCaAl_2O_3$ nanoadsorbent of crystallite size of a diameter of 55 nm to use it as a superior adsorbent material to remove the $Zn^{+2}$, $Pb^{+2}$, and $Cd^{+2}$ from polluted water. The maximum removal efficiency was achieved at pH 8, with an equilibrium time of 30 min using 1 g of the adsorbent. Afterward, the adsorption kinetics were examined via P.F.O. and PSO models; PSO gives the best fit for all adsorbents. In addition, the adsorption isotherms were considered at equilibrium. The Freundlich isotherm model clarifies the adsorption of $Zn^{+2}$, $Pb^{+2}$, and $Cd^{+2}$ by the $SnCaAl_2O_3$ nanoadsorbent. The thermodynamic studies prove that $Zn^{+2}$, $Pb^{+2}$, and $Cd^{+2}$ adsorption onto $SnCaAl_2O_3$ nanoadsorbent is exothermic and spontaneous. $SnCaAl_2O_3$ nanoadsorbent was recycled to remove the concerned adsorbent for five cycles, with high adsorption efficiency ranging from 88% to about 100%, and after the fifth regeneration cycle, adsorption efficiency reached about 88%. $SnCaAl_2O_3$ nanoadsorbent exhibited suitable selective adsorption for $Zn^{+2}$, $Pb^{+2}$, and $Cd^{+2}$ even in the presence of $Na^+$ and/or $Ca^{2+}$ competing cations. The

**Table 4. The comparison of maximum adsorption capacity of Zn$^{+2}$, Pb$^{+2}$, and Cd$^{+2}$ on various adsorbents at neutral pH (6.5–8.5).**

| Adsorbent | Metal$^{+2}$ | $q_{max}$ (mg/g) | Ref. |
|---|---|---|---|
| Soy protein | Zn | 39.780 | [82] |
| | Cd | 10.914 | |
| | Pb | 11.526 | |
| Chitosan polyitaconic acid | Cd | 36.720 | [83] |
| | Pb | 16.422 | |
| Calcium alginate | Pb | 5.304 | [84] |
| | Cd | 8.160 | |
| Succinylated starch | Zn | 0.204 | [85] |
| | Cd | 7.548 | |
| Dibenzo-18-crown-grafted corn starch | Zn | 5.712 | [86] |
| | Cd | 2.142 | |
| Cross-linked starch phosphate carbamate cross-linked starch phosphate | Pb | 2.040 | [87] |
| | Zn | 1.122 | |
| | Pb | 58.854 | |
| | Pb | 33.354 | |
| SnCaAl$_2$O$_3$ nanoparticles | Zn | 290 | this study |
| | Pb | 345 | this study |
| | Cd | 382 | this study |

prepared SnCaAl$_2$O$_3$ nanoadsorbent attributes make it a promising, low-cost, efficient adsorbent for treating wastewater.

## Supporting information

**S1 Fig. Adsorption kinetics of (a,b) Zn$^{+2}$, (c,d) Pb$^{+2}$, and (e,f) Cd$^{+2}$ adsorption on SnCaAl$_2$O$_3$ nanoadsorbent.**
(DOCX)

**S2 Fig. Adsorption isotherm for (a,b) Zn$^{+2}$, (c,d) Pb$^{+2}$, and (e,f) Cd$^{+2}$ ions.**
(DOCX)

**S3 Fig. The linear plot lnK$_c$ versus 1/T for (a) Zn$^{+2}$, (b) Pb$^{+2}$, and (c) Cd$^{+2}$ adsorption on the SnCaAl$_2$O$_3$ nanoparticles.**
(DOCX)

## Author Contributions

**Conceptualization:** Moustafa Gamal Snousy, Mahmoud F. Mubarak, Ahmed H. Ragab, Atef Mohamed Gad Mohamed, Abeer El Shahawy.

**Data curation:** Ali Sayqal, Moustafa Gamal Snousy, Mahmoud F. Mubarak, Atef Mohamed Gad Mohamed.

**Formal analysis:** Moustafa Gamal Snousy, Mahmoud F. Mubarak, Ahmed H. Ragab.

**Funding acquisition:** Ali Sayqal, Ahmed H. Ragab.

**Investigation:** Ali Sayqal, Moustafa Gamal Snousy, Mahmoud F. Mubarak, Atef Mohamed Gad Mohamed, Abeer El Shahawy.

**Methodology:** Atef Mohamed Gad Mohamed, Abeer El Shahawy.

**Supervision:** Abeer El Shahawy.

**Writing – original draft:** Ali Sayqal, Moustafa Gamal Snousy, Mahmoud F. Mubarak, Ahmed H. Ragab, Atef Mohamed Gad Mohamed, Abeer El Shahawy.

**Writing – review & editing:** Moustafa Gamal Snousy, Atef Mohamed Gad Mohamed, Abeer El Shahawy.

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
