## [Decision Letter · Decision Letter 0]

26 Sep 2022

PONE-D-22-24872Synthetization and Characterization of SnCaAl2O3 Nanocomposite as a Superior Adsorbent for Pb, Zn, and Cd Ions in Polluted WaterPLOS ONE

Dear Dr. EL Shahawy,

Thank you for submitting your manuscript to PLOS ONE. After careful consideration, we feel that it has merit but does not fully meet PLOS ONE’s publication criteria as it currently stands. Therefore, we invite you to submit a revised version of the manuscript that addresses the points raised during the review process.

We look forward to receiving your revised manuscript.

Kind regards,

Moonis Ali Khan, Ph.D.

Academic Editor

PLOS ONE

“The authors would like to express their gratitude to the Dean of Science and Research at King Khalid University for giving financial support via the General Research Project: grant no. (R.G.P.1/28/43), Saudi Arabia. And The authors would like to thank the ‎Deanship of Scientific Research at Umm Al-Qura University ‎for supporting this work by Grant Code: (22UQU4280446DSR01).”

“The Dean of Science and Research at King Khalid University via the General Research Project: grant no. (R.G.P.1/28/43). The authors would like to thank the ‎Deanship of Scientific Research at Umm Al-Qura University ‎for supporting this work by Grant Code: (22UQU4280446DSR01).”

Reviewers' comments:

Reviewer's Responses to Questions

**Comments to the Author**

1. Is the manuscript technically sound, and do the data support the conclusions?

Reviewer #1: Partly

Reviewer #2: Yes

Reviewer #3: Yes

2. Has the statistical analysis been performed appropriately and rigorously? 

Reviewer #1: N/A

Reviewer #2: Yes

Reviewer #3: Yes

3. Have the authors made all data underlying the findings in their manuscript fully available?

Reviewer #1: Yes

Reviewer #2: Yes

Reviewer #3: Yes

4. Is the manuscript presented in an intelligible fashion and written in standard English?

Reviewer #1: Yes

Reviewer #2: Yes

Reviewer #3: Yes

5. Review Comments to the Author

Reviewer #1: The work reported in this manuscript is interesting and well presented. However, it requires ‎corrections and improvements before the acceptance. The work requires revision. Some ‎comments are:‎

‎1.‎ The title can be better and more attractive; it should be more precise and represent to the ‎contents, ‎

‎2.‎ Units in all sections should be uniform, Significant figures should be uniform,‎

‎3.‎ References must be cited in the correct place in the text, and also must match correctly their ‎position in the list. Please Cite references at appropriate locations and list them correctly. Spell of ‎references must be checked.‎

‎4.‎ Ensure that all figures are cited in the text.‎

‎5.‎ The abstract is concise and accurately summarizes the essential information. Abstract should be ‎rewritten to summarize the work; the abstract should briefly state the purpose of the ‎research, the principal results, and major conclusions. An abstract is often presented ‎separately from the article, so it must be able to stand alone. Add quantitative data ‎

‎6.‎ More elaboration on the chemical interaction between the components is required.‎

‎‎7.‎ The introduction; what is the gab to cover

‎8.‎ Batch adsorption study; please add conditions in quantity how the tests were ‎performed

9. The effect of solution pH is questionable? How come the range of solution pH can be extended up to 9, how about the effect of metal ion precipitation in basic pH environment?

10. Line 257 “its maximum at pH 8 with the highest removal percent of 92.8%, 86%, and 82% for the ions of…” The highest removal at pH 8 is mainly due to precipitation effect.

11. Fig. 9 is questionable, first the range of solution pH should not exceed 7, second the Y-axis scale for R% should be in range 0-100%.

‎‎12.‎ Please improve the analysis and interpretation of Adsorption results; the following references are useful: DOI: 10.1007/s10924-021-02160-z; DOI: 10.5004/dwt.2018.21976; DOI: 10.1007/s10924-020-01734-7, DOI: 10.1016/j.ijbiomac.2021.08.160.

13.‎ Clearly indicate by numbers how many tests you did under testing: and justify why you ‎selected this

‎14.‎ Captions of the figure and tables must be with complete information, conditions etc ‎

‎15.‎ Values in the tables should be of uniform significant figures, please recheck

‎16.‎ Please improve the conclusion with clear quantitative findings ‎

‎17.‎ More emphasis on finding and its implication may be mentioned in the conclusion section.‎

‎18.‎ Typos are to be corrected, also check the equations, English must be improved .‎

‎19.‎ Add experimental conditions to captions of each figure.‎

Reviewer #2: The search for novel materials with high efficacy to remove metals from effluents is a topic of great environmental and industrial interest. The manuscript by Ali et al synthesized and characterized SnCaAl2O3, on which adsorption characteristics Zn2+, Cd2+, and Pb2+ were tested and analyzed through batch mode operation. The effects of operating parameters such as adsorbent dosage, initial concentration, pH, and temperature on the removal of Zn2+, Cd2+, and Pb2 were investigated. The manuscript could be acceptable for publication in PLOS ONE considering the following comments before a final decision:

1. Acid-based titration (Boehm’s titration) method is used to determine the number of surface oxygen groups (acid or basic) present on the carbon surface. Can the authors justify why they did not conduct Boehm’s titration?

2. It’s recommended to enrich the introduction section by especially indicating the significance of choosing Zn2+, Cd2+, and Pb2+ as mode pollutants (mode adsorbate). The reasons lying behind choosing Zn2+, Cd2+, and Pb2+ should be clearly presented.

3. More explanation about the data presented in table 4 must be inserted into the text carefully evaluating and comparing the results of the previous studies with the current ones. The author must denote what is natural Ph in table 4 so that a sound comparison with the literature data can be performed.

4. The standard deviation should be included in all numerical results, as error bars (in all Figures) or as +/- sd (text and Tables).

5. The authors should decide whether to put the graph or table for the isotherm. Putting both is redundant. One of them should be moved to the supporting information.

6. There are too many figures, and the author should combine some of them or transfer some of them to SI.

7. I am also wondering in the regeneration, how the separation of the SnCaAl2O3 was done from the water. Filtration? Centrifugation? Please clarify.

8. It’s necessary for the author to give the maximum acceptable limits for Zn2+, Cd2+, and Pb2+ respectively in both drinking as well as in wastewater. This can be added in the introductory section.

9. Please address novelty and originality in the introduction and discussion. The acceptance of the manuscript is contingent upon the incorporation of this point. Finally, there is some grammatical error in the manuscript, I strongly recommend that the language should be improved.

Reviewer #3: Manuscript ID: PONE-D-22-24872

Title: Synthetization and Characterization of SnCaAl2O3 Nanocomposite as a Superior Adsorbent for Pb, Zn, and Cd Ions in Polluted Water

Journal: PLOS ONE

SnCaAl2O3 core-shell nanoparticles (CNPs) were synthesized in the α-Alumina phase by thermal annealing of a stacked structure sandwiched between two Al 2 O 3 layers at low temperatures. The obtained structure showed Sn NP floating gate with an Al 2 O 3 dielectric stacked tunneling barrier. To characterize the prepared composites X-ray diffraction (XRD), field emission scanning electron microscope (FESEM), and high-resolution transmission electron microscopy (HR-TEM) was used. The synthesized SnCaAl 2 O 3 CNPs composite was utilized as an adsorbent for the removal of Zn, Cd, and Pb divalent cations. The removal efficiency was studied by various parameters such as adsorbent dose, pH, contact time, metals concentrations, temperature, and coexisting ions. The results are well supported by the conclusion. I recommend minor revision of the manuscript before it can be accepted. I request the authors at addressing all comments and suggestions listed below.

Comments and suggestions:

1. Abstract- “Moreover, the adsorption thermodynamic behavior of Zn+2, Cd+2, and Pb+2 30 on the synthesized composite.” – The metal ions should be written uniformly with valency state throughout the manuscript.

2. “Water pollution by toxic heavy metal ions is a critical environmental problem that may pose serious health effects.”--- I suggest the author, to discuss a paragraph related to water pollution due to presence of different contaminants and applications of different adsorbents for the treatment techniques. The authors are recommended to check the below related references, which will improve the supporting information.

Journal of Cleaner Production 241, 2019, 118263

Environmental research 2019, 170, 389-397

Journal of Molecular Liquids 317, 2020, 113916

Journal of environmental management 219, 2018, 285-293

3. The author mention, “Adsorption of ions on the surface of many solid materials such as clay, zeolites, activated charcoal, or silica gel is the common removal system” – The statement needs supporting citations as well.

Journal of Hazardous Materials 400, 2020, 123247

Desalination and Water Treatment 57 (46), 2016, 21863-21869

Nanomaterials 9 (5), 2019, 776

Microporous and Mesoporous Materials 261, 2018, 198-206

4. “A known amount of the as-prepared nano Al2O3 shells was first dispersed in 20 mL of deionized water, followed by a pre-determined amount of CTAB, CaCl2, and SnCl2 with stirring and heating at 300 rpm at 40 °C for 2 h, respectively”-Does the author followed any reported literature?

5. The author need to discuss the regeneration methods of the adsorbent materials.

6. The English quality not up to the mark. All the typos and grammar need to check thoroughly in the manuscript.

7. “Figure 10. The dose effect on the SnCaAl2O3 nanoparticles adsorptions.”—Add the optimal conditions in each caption.

6. PLOS authors have the option to publish the peer review history of their article (what does this mean?). If published, this will include your full peer review and any attached files.

Reviewer #1: No

Reviewer #2: No

Reviewer #3: No

---

## [Decision Letter · Decision Letter 1]

17 Oct 2022

Synthetization and Characterization of SnCaAl2O3 Nanocomposite and Using as a Superior Adsorbent for Pb, Zn, and Cd Ions in Polluted Water

PONE-D-22-24872R1

Dear Dr. EL Shahawy,

We’re pleased to inform you that your manuscript has been judged scientifically suitable for publication and will be formally accepted for publication once it meets all outstanding technical requirements.

Kind regards,

Moonis Ali Khan, Ph.D.

Academic Editor

PLOS ONE

Additional Editor Comments (optional):

Reviewers' comments:

Reviewer's Responses to Questions

**Comments to the Author**

1. If the authors have adequately addressed your comments raised in a previous round of review and you feel that this manuscript is now acceptable for publication, you may indicate that here to bypass the “Comments to the Author” section, enter your conflict of interest statement in the “Confidential to Editor” section, and submit your "Accept" recommendation.

Reviewer #1: All comments have been addressed

2. Is the manuscript technically sound, and do the data support the conclusions?

Reviewer #1: Yes

3. Has the statistical analysis been performed appropriately and rigorously? 

Reviewer #1: N/A

4. Have the authors made all data underlying the findings in their manuscript fully available?

Reviewer #1: Yes

5. Is the manuscript presented in an intelligible fashion and written in standard English?

Reviewer #1: Yes

6. Review Comments to the Author

Reviewer #1: The authors did carefully all the required corrections , and the revised version is publishable in current form

7. PLOS authors have the option to publish the peer review history of their article (what does this mean?). If published, this will include your full peer review and any attached files.

Reviewer #1: No

---

## [Editor Report · Acceptance letter]

24 Oct 2022

PONE-D-22-24872R1 

Synthetization and Characterization of SnCaAl_2_O_3_ Nanocomposite and Using as a Superior Adsorbent for Pb, Zn, and Cd Ions in Polluted Water 

Dear Dr. El Shahawy:

I'm pleased to inform you that your manuscript has been deemed suitable for publication in PLOS ONE. Congratulations! Your manuscript is now with our production department. 

Kind regards, 

on behalf of

Dr. Moonis Ali Khan 

Academic Editor

PLOS ONE